# Physiologically Based Pharmacokinetic Modelling of Serum 25-Hydroxyvitamin D Concentrations in Schoolchildren Receiving Weekly Oral Vitamin D_3_ Supplementation

**DOI:** 10.3390/nu17193028

**Published:** 2025-09-23

**Authors:** Nadda Muhamad, Neil Walker, Keren Middelkoop, Davaasambuu Ganmaa, Adrian R. Martineau, Tao You

**Affiliations:** 1Department of Pharmacology and Therapeutics, University of Liverpool, Liverpool L69 7BE, UK; muhamad_n@su.ac.th; 2Centre of Excellence in Long-Acting Therapeutics (CELT), University of Liverpool, Liverpool L69 7BE, UK; 3Department of Biomedicine and Health Informatics, Faculty of Pharmacy, Silpakorn University, Nakhon Pathom 73000, Thailand; 4Blizard Institute, Faculty of Medicine and Dentistry, Queen Mary University of London, London E1 2AT, UK; neil.walker@qmul.ac.uk (N.W.); a.martineau@qmul.ac.uk (A.R.M.); 5Institute of Infectious Disease and Molecular Medicine, University of Cape Town, Cape Town 7925, South Africa; keren.middelkoop@hiv-research.org.za; 6Desmond Tutu HIV Centre, Department of Medicine, University of Cape Town, Cape Town 7925, South Africa; 7Department of Nutrition, Harvard School of Public Health, Boston, MA 02115, USA; gdavaasa@hsph.harvard.edu; 8Beyond Consulting Ltd., 14 Tytherington Park Road, Macclesfield, Cheshire SK10 2EL, UK

**Keywords:** vitamin D, healthy children, PBPK modelling, nonlinear mixed effects modelling

## Abstract

**Background:** Following vitamin D_3_ oral administration, attained serum concentrations of its metabolite 25-hydroxyvitamin D_3_ (25(OH)D_3_) are variable among children. **Methods:** We developed physiologically based pharmacokinetic (PBPK) modelling using annually measured serum 25(OH)D_3_ concentrations in 77 Cape Town schoolchildren aged 6–11 years who received weekly oral doses of 10,000 IU vitamin D_3_ for 3 years during a clinical trial (Δ25(OH)D = 32.2 nmol/L, 95% CI: [−3.2, 65.8] nmol/L). Simulations were performed to test the model on 463 other participants in the same trial, and in a cohort of 1756 Mongolian schoolchildren aged 6–11 years who received weekly oral doses of 14,000 IU vitamin D_3_ for 3 years in another trial. **Results:** The best model attributed most of the variability in post-supplementation 25(OH)D_3_ concentrations to hepatic clearance and covariates including weight (ΔAIC = −21) and ZBMI (body mass index Z-score, ΔAIC = −34). For 463 other children from the Cape Town trial (Δ25(OH)D = 25.8 nmol/L, 95% CI: [8.3, 47.2] nmol/L), mean estimation error was 5.3 nmol/L, and 76.7% of observations were within the 95% prediction intervals. Our simulation supported the previous proposal that serum 25(OH)D_3_ should exceed 50 nmol/L among 97.5% of European children at 24.4 μg/day vitamin D_3_ dosing. At a higher weekly dose (14,000 IU), the Mongolian children demonstrated a higher average increase in serum 25(OH)D_3_ (40.6 [−2.9, 88.9] nmol/L) but were overestimated by the model. **Conclusion:** We developed the first PBPK model to successfully predict the long-term serum 25(OH)D_3_ increases in healthy schoolchildren in Cape Town who received orally administered vitamin D_3_ and exhibited higher relative increases than Mongolian children.

## 1. Introduction

Vitamin D plays crucial roles in bone health and development in children, as well as in maintaining calcium, phosphate, and magnesium homeostasis in the body [1,2]. Vitamin D deficiency is defined as serum 25-hydroxyvitamin D_3_ (25(OH)D_3_) concentrations being less than 50 nmol/L by the Institute of Medicine (IOM) [3]. Following this definition, a worldwide study spanning 2000 to 2022 revealed 48.5% (90% CI: [42.5–54.5%]) of individuals < 18 years of age are deficient [4]. In particular, regional studies found deficiency in 39.1% of under 18-year-olds in the United Arab Emirates [5], 64% of the 9 to 13-year-olds in Poland in March [6], and 65% of children aged 6 to 12 years in Pakistan [7]. Deficiency is partly due to insufficient sun exposure during the winter and spring [5,6,7,8]. By even the most conservative serum 25(OH)D_3_ threshold of 25 nmol/L, vitamin D deficiency is still frequent in both low- and high-income countries [8].

A meta-regression of 1429 subjects across 11 trials with vitamin D-fortified foods, conducted by Cashman and colleagues [9], revealed that the dietary requirement for 97.5% of children aged 2–17 years to maintain serum 25(OH)D_3_ over 50 nmol/L is 24.4 µg/day (95% CI: 16.8–38.2 µg/day) [9]. This model was adjusted for baseline levels, age, and BMI from the population living between latitudes 40° and 63° N [9].

Compared with regression modelling, pharmacokinetic (PK) modelling is often used to predict longitudinal changes in circulating concentrations of metabolites following administration of a drug or nutritional supplement, and has been adopted to inform dosing guidelines for paediatric clinical care [10,11]. Hence, there is a need to develop a PK model to evaluate the proposed 24.4 µg/day dose in healthy children.

Previously, a one-compartment model of 25(OH)D_3_ PK in children with chronic kidney disease suggested that a weight-based dosing strategy is needed [12]. Unfortunately, population predictions using this model systematically overestimated serum 25(OH)D_3_ concentrations, indicating that the model structure needed changing in order to make reliable predictions.

For healthy adults, our previous physiologically based pharmacokinetic (PBPK) modelling accurately predicted the mean serum 25(OH)D_3_ concentrations across a wide range of vitamin D_3_ doses in 155 trial arms, from 12.5 µg/day to 1250 µg/day for daily dosing, and up to 50,000 µg for bolus dosing [13]. This naïve average model highlighted the nonlinear dose–exposure relationship for 25(OH)D_3_. We further developed the model into a nonlinear mixed effects model and identified weight as a predictive covariate for PK in healthy adults [14].

Two randomised controlled trials (ViDiKids trials) were conducted to evaluate the effects of vitamin D supplementation in healthy children in Cape Town, South Africa [15], and Ulaanbaatar, Mongolia [16]. Participants received weekly vitamin D doses, and these data provide a starting point for constructing PK models for healthy children.

Here, we hypothesise that children mainly differ from adults in the disposition and clearance of 25(OH)D_3_. To test this, we changed the physiological parameters for children, fitted only these two parameters, and fixed all other parameters from the adult model [13] to see if the model can fit the data. The model successfully fitted the data from the ViDiKids Cape Town trial.

We then used this PK model to examine the 24.4 µg/day dose recommended for healthy children [9]. Secondly, we used the model to explore what might underpin the PK differences between Cape Town and Mongolian children.

## 2. Materials and Methods

### 2.1. Study Participants and Clinical Data Collection

#### 2.1.1. Cape Town Participants

Children in Grades 1–4 (aged 6 to 11 years) at participating schools in the Klipfontein district of Cape Town, South Africa, were recruited to the ViDiKids trial [15]. Over a 3-year period, each child, randomised to the intervention arm, received a weekly oral capsule containing 250 µg (10,000 IU) of vitamin D_3_. Capsule administration was recorded by the trial team during term-times and was supervised by parents/guardians during school holidays and COVID-19 lockdowns. Dosing records for each participant were used for model fitting and simulation. All received doses were assumed to be taken. Blood samples were collected at baseline and 36 months for all participants, and at 6, 12 and 24 months in a sub-set. Serum 25(OH)D_2_ and 25(OH)D_3_ concentrations were quantified with liquid chromatography tandem mass spectrometry [17]. The concentration of serum 25(OH)D_2_ was comparable to approximately 1% of serum 25(OH)D_3_, and serum 25(OH)D_2_ was not consistently detected. The fitting and testing of the model only used the 25(OH)D_3_ data. For simplicity, vitamin D_3_ and 25(OH)D_3_ are referred to as vitamin D and 25(OH)D onwards, respectively. In addition, the 6-month time point was confounded by seasonal changes and was excluded from modelling. The model was fitted to 77 children and was tested on data from 463 other children.

#### 2.1.2. Ulaanbaatar Participants

Schoolchildren aged 6 to 14 years, from participating schools in Ulaanbaatar, Mongolia, were recruited to the randomised controlled trials of vitamin D supplementation to prevent tuberculosis infection from September 2015 to March 2017. The CONSORT diagram is provided in Appendix A. Over the 3-year study period, each child, randomised to the intervention arm, received a weekly oral capsule containing 350 µg (14,000 IU) vitamin D_3_ or a placebo. Blood samples were collected at baseline and at the end of the third year. Serum 25(OH)D_3_ concentrations were determined using an enzyme-linked fluorescent assay [16]. The study was not interrupted by major breaks. Compliance was recorded. The treated children whose serum 25(OH)D at baseline and follow-up were above the lower limit of quantification (14.2 nmol/L) were simulated, using the Cape Town model with or without modifications.

### 2.2. Physiological Parameters

We averaged the cardiac output for boys (4.8 L/min) and girls (4.3 L/min) aged 7 to 9 [18] and took the mean 4.55 L/min (273 L/h) for our model. The average blood flow for children in this model was estimated assuming the fraction of blood flow going into each compartment was the same as a 70 kg man [19]. For simplicity, we assumed the proportion of each compartment’s volume was the same for children and adults, with a 1 kg/L apparent density. All physiological parameters are presented in Appendix A.

### 2.3. Model Structure

We developed a series of 11 nested models with different covariate structures to select the best PBPK model.

The base model (Model 1) was based on our published adult model [13], which considered the arterial blood (V_art_), venous blood (V_ven_), the liver (V_l_), and the rest of the body compartment (V_rb_), which lumped together all non-elimination organs. These four compartments were applied to models 1–8 and 11 (Appendix A). In models 9 and 10, non-elimination organs were divided into fat mass (V_fm_) and lean mass (V_lm_) (Appendix A). All model equations are listed in the Appendix A.

Partition coefficients K_p_ were introduced to describe the ratio between specific compartments and the venous blood. K_p_ > 1 indicates that the concentration is higher in a compartment than in the venous blood. K_prb_ and K_p25rb_ are for vitamin D and 25(OH)D in the rest of the body compartment, respectively; K_pl_ and K_p25l_ are for vitamin D and 25(OH)D in the liver, respectively; K_pfm_ is for 25(OH)D in fat mass (models 9–10); and K_plm_ is for 25(OH)D in lean mass (models 9–10).

F_m_ refers to the fraction of vitamin D metabolised into 25(OH)D, which was assumed to be 0.33 [13]. CL_H_ is the hepatic clearance rate (L/h).

The clearance of 25(OH)D was described by a sigmoidal function, to describe the nonlinear relationship between dose and 25(OH)D serum concentration:$$\frac{{\mathrm{CL}}_{\max}\times {C_{25\left(OH\right)D}}^{\gamma }}{{C_{50}}^{\gamma }+{C_{25\left(OH\right)D}}^{\gamma }}\times C_{25\left(\mathrm{OH}\right)D}$$

To speed up simulation, we assumed vitamin D level in venous blood was kept in a steady state with arterial blood and the rest of the body. Thus, vitamin D concentrations of these two compartments were explicit functions of the venous blood concentration (Equations (1) and (2)). Simulated 25(OH)D was not affected by this assumption and the computing time using the “daspk” ordinary differential equations (ODE) solver, an implicit method for stiff ODEs and differential-algebraic equations (DAEs) implemented in R version 4.2.2, was reduced by ~90%. See Appendix A for details.(1)$$A_{\mathrm{art}}=\frac{A_{\mathrm{ven}}}{V_{\mathrm{ven}}}\times \frac{Q_{\mathrm{co}}}{(Q_{l}+Q_{\mathrm{rb}})\times V_{\mathrm{art}}}$$(2)$$A_{\mathrm{rb}}=V_{\mathrm{rb}}\times \frac{K_{\mathrm{prb}}}{V_{\mathrm{art}}\times A_{\mathrm{art}}}$$

A_art_ and A_ven_ are the amount of vitamin D in the arterial and venous blood, respectively. Q_co_, Q_l_, and Q_rb_ are the cardiac output, the liver blood flow, and the blood flow to the rest of the body, respectively. V_art_ and V_ven_ are the volume of distribution for the arterial and venous blood, respectively.

All models assumed combined additive and proportional residual errors. The fixed and random effects are tabulated (Table 1 and Appendix A).

The base model (Model 1) did not include any covariate to estimate parameters. All other models assumed that the volume of distribution of each compartment was directly proportional to weight (WT), and the maximum clearance rate for serum 25(OH)D (CL_MAX_) was proportional to WT^0.75^. Models 2–4 and 11 fixed the weight to its baseline value for each child. Models 5 and 6 assumed linear weight gain:(3)WT(t) = WT(0) + 3.08 kg/year × time (year) for boys(4)WT(t)=WT(0)+3.23 kg/year × time (year) for girls

Models 7–10 interpolated weight from baseline to 3 years linearly for each child:(5)$$\mathrm{WT}(t)\;=\;\mathrm{WT}(0)\;+\;[\mathrm{WT}(3y)\;-\;\mathrm{WT}(0)]\;\mathrm{kg}\;\times \;\frac{\mathrm{time}\;(\mathrm{year})}{3\;\mathrm{years}}$$

In Model 6, $K_{p25\mathrm{rb}}$ was related to the BMI-for-age Z-score (ZBMI) via $K_{p25\mathrm{rb}}^{ZBMI}$:(6)$$K_{p25\mathrm{rb}}=e^{{\mathrm{TK}}_{p25\mathrm{rb}}+\mathrm{ZBMI}\times K_{p25\mathrm{rb}}^{ZBMI}+\eta_{K_{p25\mathrm{rb}}}}$$

In Models 9 and 10, the fraction of fat mass was reported to be related to ZBMI [20]:(7)$$f_{\mathrm{FM}}=\left\{\begin{array}{c} max(\frac{28.61+7.82\times \mathrm{ZBMI}-0.91\times {\mathrm{ZBMI}}^{2}+0.03\times {\mathrm{ZBMI}}^{3}}{100},0.05),\;if\;\mathrm{ZBMI}\leq 6.19\\ 0.49,\;if\;\mathrm{ZBMI}>6.19 \end{array}\right.$$

### 2.4. Model Assessment and Selection

All nonlinear mixed-effect (NLME) models were fitted using the stochastic approximation expectation-maximisation (SAEM) routine in the nlmixr2 R package. To assess the fitting, we visualised the goodness-of-fit, distribution of conditional weighted residuals (CWRES), normalised prediction distribution errors (NPDEs), and performed a visual predictive check (VPC). We deemed *η* shrinkage < 30% to be acceptable. Akaike information criterion (AIC) and Bayesian information criterion (BIC) were assessed for model selection. As conclusions based on the AIC and BIC were consistent, we did not prioritise one over the other.

For VPC, 1000 samples of random effects ($\eta_{{\mathrm{Kp}}_{25\mathrm{rb}}}$ and $\eta_{{\mathrm{CL}}_{\mathrm{MAX}}}$) and residual errors were drawn to simulate the 5th, 50th, and 95th percentiles of serum 25(OH)D, together with 90% prediction intervals (PI) around each percentile.

### 2.5. Simulations

To test the model independently, we simulated 9 children in the treatment arm of the safety sub-study (baseline, 1, and 2 years) and another 454 children in the treatment group (baseline and 3 years). For each child, we generated 30 virtual studies to estimate the 95% prediction intervals of the simulated 2.5th, 50th, and 97.5th percentiles. In each study, a realisation of the fixed effect uncertainty (${\mathrm{CL}}_{\max}$ and $K_{p25\mathrm{fm}}$) was made, and 2500 samples of the IIV ($\eta_{{\mathrm{CL}}_{\max}}$) and the additive residual error were generated to simulate the PK according to the dosing record of that child. The proportional residual error was not sampled to ensure simulations started with the observed baseline values.

To simulate the European children, a virtual population was generated for 1250 boys and 1250 girls for 6-year-olds and 11-year-olds, respectively. Baseline serum 25(OH)D was sampled as per the reported 5th, 25th, 50th, 75th, and 95th percentiles for European children [21]. For simplicity, points were sampled uniformly between each pair of neighbouring percentiles. To cover the whole range, we assumed the lower and upper bounds of 10 and 90 nmol/L, respectively. The WHO provides weight-for-age information from birth to 10 years. For the 6-year group, weight was sampled as per the 1st, 3rd, 5th, 15th, 25th, 50th, 75th, 85th, 95th, 97th, and 99th percentiles reported by the WHO [22]. Similarly, uniform distribution was assumed between two percentiles. We used the weight-for-age Z-score as a proxy for the BMI Z-score. For the 11-year-old group, the WHO does not provide weight distribution, but instead reports BMI and height distribution. Hence, we initially sampled the body mass index-for-age [22] and height-for-age [23] percentiles (both at 1st, 3rd, 5th, 15th, 25th, 50th, 75th, 85th, 95th, 97th, and 99th), assuming independent distribution, and calculated the weight for each virtual child. As was carried out for the simulation of each child, 30 virtual studies were generated to sample the fixed effect uncertainty, random effects, and the additive errors to estimate the 95% prediction intervals of the simulated 2.5th, 50th, and 97.5th percentiles.

### 2.6. Comparing Trial Results

To compare the results from the Cape Town trial with its sister Mongolian trial, we compared the mean weight and serum 25(OH)D at baseline, weekly dose (250 µg Cape Town v 350 µg Mongolia), compliance (i.e., received doses/number of weeks × 100%), and serum 25(OH)D increase. We graphed the 3-year changes in serum 25(OH)D with compliance. Baseline weight and age were fitted to a linear regression model to infer the mean weight increase each year (boy: 3.0 kg/year; girl: 3.4 kg/year). Similarly to the European children described in Section 2.5, we simulated the Mongolian children where each child started with their baseline weight and baseline serum 25(OH)D and was assumed to conform to the mean weight increase per year. Apart from this, simulation was also carried out at $4\times {CL}_{max}$.

### 2.7. Computing Software and Environment

Baseline characteristics were analysed descriptively using IBM SPSS software, version 25 (IBM, Armonk, NY, USA). Categorical variables were summarised as frequencies and percentages, while continuous variables were presented as means with standard deviations. ZBMI for each child was computed in the WHO 2007 Z-Score calculator shiny app [24]. For NLME fitting, we used the nlmixr2 package version 2.0.9 in R (version 4.2.2) on a Linux high-performance cluster with 60 cores. Model fitting was performed using the SAEM routine implemented in nlmixr2. VPC was carried out using vpcSim function from the nlmixr2est R package version 2.1.5. Simulations were carried out with the rxode2 R package version 2.0.13 [25]. Graphical diagnostics were conducted using RStudio version 4.3.1.

## 3. Results

Out of the 1682 participants in the ViDiKids trial, the first 200 participants enrolled in the study were also enrolled in a safety study. Among the 94 children in the treatment arm of this safety study, 77 children had their serum 25(OH)D consistently assessed at baseline, 1, 2, and 3 years, and were used to fit the model; 9 were assessed at baseline, 1 and 2 years, and were used to test the model; 8 had missing values and were excluded (Figure 1) [17]. Another 454 children were assessed at baseline and 3 years. They were also used to test the model (Figure 1). The baseline age and serum 25(OH)D of the 77 children for fitting were 8.9 ± 1.3 years and 64.7 ± 14.9 nmol/L, respectively. The baseline data were collected between March and September in 2017 (Appendix A).

In the 94 children in the intervention arm of the safety study, mean serum 25(OH)D showed statistically significant increases (Appendix A). Baseline 25(OH)D was somewhat anti-correlated with ZBMI among the 77 children for model fitting (slope = −2.644 nmol/L; *p*-value = 0.10) (Appendix A).

### 3.1. The Final Model

The final model encompasses five compartments for vitamin D, including the GI tract, arterial blood, venous blood, the liver, and the rest of the body, which lumps together all non-eliminating organs. For 25(OH)D, non-eliminating organs were divided into fat mass and lean mass (Figure 2). Predictive covariates include weight (to estimate the volume of distribution) and ZBMI (to estimate the fat mass and lean mass) [26]. Similarly to our previous healthy adult PBPK model [13], a combined basal rate of vitamin D cutaneous synthesis and intake in each child was calculated so that the model was in balance with the baseline 25(OH)D prior to dosing (see Appendix A for details).

Model fitting made good parametric inference, with η shrinkage < 30% (Table 2). Additive error (0.00249 nmol/L) was negligible, and proportional residual error was 0.109, which was small (Table 2). The random effect $\eta_{{CL}_{max}}$ was independent from covariates (Appendix A), and the model was fitted at the optimal value of Kp_25lm_ = 4 (Appendix A).

The final model successfully repeated the observed serum 25(OH)D in 77 children for 3 years. Population predictions are in good agreement with observed serum 25(OH)D (Figure 3A). The concordance with data is higher for individual predictions (R^2^ = 0.70 in Figure 3B). The CWRES and NPDEs did not exhibit systemic errors, suggesting the model predictions were unbiased (Figure 3C–F). Population and individual predictions for each child were plotted together with doses received during the 3-year span (Appendix A).

### 3.2. Visual Predictive Check

VPC verified model predictions. The observed 5th and 95th percentiles (red dashed lines in Figure 4) were within the 90% prediction intervals for the predicted 5th and 95th percentiles (shaded areas in Figure 4). The observed medians (black solid line in Figure 4) were also within 90% prediction intervals for the predicted medians. This confirms that model predictions were in good agreement with the fitting data.

### 3.3. Simulation of 463 Cape Town Children

To test the model, we first simulated 9 children from the same trial arm, who were not fitted. Here, we observed that 5 out of 9 children were entirely within the 2.5th–97.5th percentile range (Figure 5A,D,F,H,I). For the other 4 children, a 1-year observation (Figure 5C) and a 2-year observation (Figure 5E) were within the 2.5th–97.5th predicted percentiles, and observations outside the 2.5th–97.5th predicted percentiles were within ±20 nmol/L of the range (Figure 5B,C,E,G). This provided confidence in model predictions for up to 2 years of dosing.

In addition, we simulated a further 454 children from the same trial and found 76.9% of observed serum 25(OH)D were between the predicted 2.5th–97.5th percentiles (Appendix A). The mean difference between observation and the predicted 50th percentile is 5.4 nmol/L (90% CI: [−26.2, 39.5] nmol/L, Appendix A). This suggests a possible minor tendency for overprediction. Consistent with this, the training data exhibited an average larger increase (Δ25(OH)D = 32.2 nmol/L, 95% CI: [−3.2, 65.8] nmol/L) than the 463 children for testing Δ25(OH)D = 25.8 nmol/L (95% CI: [8.3, 47.2] nmol/L). The range of the predicted 2.5th to 97.5th percentiles changed over time. The range was smaller at the end of year 1, compared with the end of years 2 and 3 (Appendix A). Such a range exhibited strong anti-correlation with baseline weight (R^2^ = 0.56) and was somewhat anti-correlated with baseline ZBMI (R^2^ = 0.18) (Appendix A).

### 3.4. Simulate Cashman’s 24.4 µg Daily Dose

We simulated the proposed 24.4 µg daily dose in European children of 6 and 11 years of age (Figure 6). Baseline 25(OH)D were sampled according to a reported European study [21]. Weight was based on WHO reports [22]. At 100% compliance, over 97.5% of the 6-year-old children were predicted to reach the 50 nmol/L threshold within 21 weeks (Figure 6A). For the 11-year-old children, 97.5% were predicted to reach the same threshold around 39 weeks (Figure 6B). This is intuitive, as older children are heavier. Their larger volume of distribution reduces the increase in serum 25(OH)D. In short, model simulation results were consistent with the expectation that 24.4 µg daily vitamin D supplementation would be sufficient to reach the 50 nmol/L threshold among children aged 6–11 years.

### 3.5. Comparing Trials in Cape Town and Ulaanbaatar

Finally, we compared the Cape Town trial with the sister Mongolian trial (Appendix A). The mean baseline weight was similar for both studies (Appendix A), but baseline serum 25(OH)D was higher among the Cape Town children (mean = 64 nmol/L) than Mongolian children (mean = 38 nmol/L). The Mongolian trial had higher compliance (Appendix A) and used a higher weekly dose (350 µg Mongolia v 250 µg Cape Town) than the Cape Town trial. The average change in serum 25(OH)D was higher among the Mongolian children (Appendix A: 40.6 nmol/L, 95% CI = [−2.9, 88.9] nmol/L) than the Cape Town children (Appendix A: 32.2nmol/L, 95% CI: [−3.2, 65.8] nmol/L). The 3-year changes in serum 25(OH)D did not appear to strongly correlate with compliance (Appendix A).

Our model marginally overpredicted the 463 Cape Town children by 5.4 nmol/L on average. However, model simulations significantly overestimated the increase in Mongolian children (Appendix A). To test whether this might be attributed to a higher clearance rate among the Mongolian children alone, we increased ${CL}_{max}$ to $0.048\;h^{-1}$ (i.e., 4-fold). Simulation still overestimated the final 25(OH)D, albeit to a lesser extent (Appendix A). This suggests other factors warrant considerations. To evaluate the effects of $C_{50}$ (25(OH)D concentration at 50% maximum clearance rate), we simulated the serum 25(OH)D in an average Cape Town child at the end of 3 years at different values of $C_{50}$: baseline weight = 30 kg, baseline 25(OH)D = 50 nmol/L, 250 µg/week, 100% compliance. We found the final 25(OH)D prediction was sensitive to $C_{50}\geq 30\;\mathrm{nmol}/L$ but the relationship was not monotonic (Appendix A: ${CL}_{max}=0.012\;h^{-1}$; Appendix A ${CL}_{max}=0.048\;h^{-1}$). Consistently, changing $C_{50}$ alone does not fit with the Mongolian data (Appendix A: $C_{50}=30\;\mathrm{nmol}/L$; Appendix A: $C_{50}=10\;\mathrm{nmol}/L$). At ${CL}_{max}=0.048\;h^{-1}$ and $C_{50}=30$ nmol/L, model simulations seem to agree with the data (Appendix A). To appreciate these, we graphed the predicted 25(OH)D clearance rate v 25(OH)D (Appendix A): Cape Town children appeared flatter than healthy adults (${CL}_{max}=0.0463\;h^{-1}$ [14]), while the dashed line for ${CL}_{max}=0.048\;h^{-1}$ and $C_{50}=30$ nmol/L was very different. This is discussed in Section 4.4.

## 4. Discussion

Currently, there is no vitamin D PK model for healthy children. Here, we reported a PBPK model for serum 25(OH)D based on 3-year observations among 77 Cape Town children. The model gave reasonable predictions for another 463 children among the same trial, for up to 3 years, with 95% of CI covering 77.3% of the observed values.

The model was built based on a parsimonious PBPK model we previously developed for healthy adults [13,14], which was constructed in methods different from other models in two main aspects: first, the baseline serum 25(OH)D was used to estimate the vitamin D intake and synthesis in each individual. Second, a nonlinear clearance term was used. These two features successfully reconciled the differences in PK for vitamin D doses across 4 orders of magnitude [13,14]. In comparison, an empirical 1-compartment PK model was developed based on 3 dosing regimens (3000 IU daily, 25,000 IU weekly, or 100,000 IU monthly for 3 months) that were roughly equivalent to each other and assumed the rate of clearance is directly proportional to serum 25(OH)D [12]. The direct consequence of the linear clearance assumption is PK should be directly proportional to the dose. This is invalid for vitamin D [13]. In addition, we considered the rapid growth in body weight among children, which is needed to fit the long-term data we used.

The model is concerned with the long-term changes in serum 25(OH)D. We identified both weight and ZMBI as covariates to predict the volume of distribution. In addition, the range of the 2.5th to 97.5th predicted percentiles at 3 years was strongly inversely correlated with weight (Appendix A). The range was also somewhat anti-correlated with ZBMI (Appendix A), probably due to the fact that weight and ZBMI were associated. This is intuitive: the narrower range for a heavy child might be due to the lipophilic nature of vitamin D, which tends to accumulate in adipose tissue, thereby reducing variability in serum levels.

Our simulation of the European children who received 24.4 µg daily vitamin D was consistent with the finding by Cashman that this dose would be sufficient to bring the serum 25(OH)D above the 50 nmol/L threshold in the majority of European children. Here are some additional considerations for model parameter and potential limitations.

### 4.1. 25(OH)D Disposition in Children

Previously, population PK modelling of children with chronic kidney disease reported the volume of distribution (V/F) for 25(OH)D of 292 L (95% CI: [199, 572] L) for a 24 kg child [12]. They used a 1-compartment model to link vitamin D_3_ dose directly to serum 25(OH)D, assuming each molecule of vitamin D_3_ produces 1 molecule of 25(OH)D. Our model assumes that every 3 molecules of vitamin D produce 1 molecule of 25(OH)D, which is more accurate. Had Wan et al. made the same assumption, their V/F would have been 97 L for a 24 kg child, or 122 L for a 30 kg child. These values would agree with our parameter values given the reasoning below.

In our PBPK model, volume of distribution V_d_ = Σ(V_organ_ × K_p_). For a 30 kg child with ZBMI = 0, 28.61% of weight is expected to be fat mass (i.e., 8.58 kg) [20]. Assuming 0.6 L arterial blood, 1.80 L venous blood, and a 0.77 L liver (with K_p_ = 1) (Appendix A), the lean mass should be 30 × (100% − 28.61%) − 1.80 − 0.77 − 0.6 = 18.25 kg. Hence, V_d_ = Σ(V_organ_ × K_p_) = 1.8 + 0.6 + 0.77 × 1 + 8.58 × 4 + 18.25 × 4.71 = 123 L for 25(OH)D. This is consistent with what was reported for children with chronic kidney disease. We speculate the high K_p_ for 25(OH)D in lean mass (K_p25lm_ = 4) may indicate that 25(OH)D efficiently binds with vitamin D receptors in children to meet the demand of rapid growth. This leads to the high V_d_ in children. In contrast, we previously reported the volume of distribution for vitamin D was between 10.0 and 16.6 L for a 70 kg man from a non-compartmental analysis [14]. In that model, the volume of distribution for the rest of the body (including all non-eliminating organs) V_rb_ = 62.2 L. Because V_d_ = Σ(V_organ_ × K_p_), we reasoned the partition coefficient for vitamin D K_prb_ must be lower than 1 [14]. Since 25(OH)D has similar physicochemical properties with vitamin D, K_p25rb_ < 1 for adults was also expected.

### 4.2. Maximum Serum 25(OH)D Clearance Rate Constant

Here, we reported that the maximum 25(OH)D clearance rate constant CL_max_ for a 30 kg child is 0.0119 h^−1^ (95% CI: [0.0102, 0.0139] h^−1^) (Table 1). For a 70 kg child, this extrapolates to 0.0225 h^−1^ (95% CI: [0.0193, 0.0262] h^−1^), and is smaller than a 70 kg adult, as we reported earlier (0.0463 h^−1^, 95% CI: [0.0359, 0.0597] h^−1^) [14]. This suggests there might be genuine differences in clearance between children and adults.

### 4.3. Limitation of the Data Used for Fitting and Testing

The data we fitted were sampled once a year for 3 years. The sampling was sparse, and we did not have observations for PK changes in shorter terms such as once every week. Hence, although the predictions made in years were reliable, the model does not necessarily capture short-term kinetics.

Secondly, the recorded dispensed doses were used to fit the model. Following long gaps in dosing (school holidays, COVID-19 lockdowns), serum 25(OH)D was predicted to diminish. Graphing the dosing information together with model predictions helped make sense of the fitting data (Appendix A). However, there were cases we could not explain, such as the lack of increase in the 1-year observations in children 41 and 92, and the 2-year observations in children 36 and 164 (Appendix A). These anomalous cases warrant further investigation.

Thirdly, the model does not consider seasonal variations. We fitted the serum 25(OH)D in the control group of the ViDiKids safety study (n = 105) to a sinusoidal curve and inferred an amplitude of 11.5 nmol/L (Appendix A). Therefore, significant fluctuation is expected for Cape Town children within a year.

Fourthly, vitamin D was not measured for the Cape Town children, and the model was not calibrated for serum vitamin D.

Lastly, our model does not consider the active metabolite 1,25-dihydroxyvitamin D (1,25(OH)_2_D). It can be further developed for children with renal impairment to evaluate the impact of renal disease on vitamin D supplementation strategies. This would invoke modifications in renal clearance and inclusion of metabolic enzymes, such as CYP27B1, which are responsible for producing 1,25(OH)_2_D in the kidney [27].

### 4.4. How Does Race Affect Serum 25(OH)D?

Vitamin D is regulated via a homeostatic mechanism that resists increases in serum 25(OH)D. An accidental overdose at 50 mg (i.e., 2,000,000 IU) in a Caucasian 90-year-old man and a 95-year-old woman resulted in an increase in serum 25(OH)D from baselines below 80 nmol/L to 527 and 422 nmol/L at 8 days after intake, respectively [28]. Although the absolute increases in serum 25(OH)D in these two cases were large, the increases were less than what is proportional to the dose. This was successfully modelled by the assumption that the increase in serum 25(OH)D was related to disproportionally higher increase in its clearance, resulting in less than proportional increase in serum 25(OH)D [13,14]. Our Cape Town children model inherits this assumption (Appendix A). The Cape Town trial used only one dose. Fitting ${CL}_{max}$, $C_{50}$ and γ in the clearance function simultaneously requires data for different doses [13]. Here, we made a practical choice of fitting ${CL}_{max}$ while fixing $C_{50}$ and γ at the adult values.

Humans adapt to various environmental conditions on earth; sun exposure and dietary intake are different. The homeostatic mechanism is speculated to have evolved differently. In the US, the level of African ancestry is found to be negatively correlated with serum 25(OH)D levels: every 10% increase in African ancestry is correlated with approximately 2.5 nmol/L reduction in serum 25(OH)D baseline levels on average [29]. The same study also estimated both sunlight and diet were 46% less effective in raising the 25(OH)D levels of participants with high African ancestry, as compared to those with low African ancestry [29].

Adaptation to low serum 25(OH)D has been shown by several studies of high-latitude populations, mainly the Inuit of Canada and Greenland but also indigenous populations of northern Asia [30,31]. Like the Inuit, Mongolians have adapted to an environment where the skin cannot produce sufficient vitamin D for most of the year.

Modelling these racial differences in 25(OH)D homeostasis requires refitting ${CL}_{max}$, $C_{50}$, and γ simultaneously, which needs data for multiple doses [13]. The Mongolian study had only two observations (baseline and final) and cannot be used for model fitting (at least 3 time points needed). Hence, a future study of different doses is required to model the Mongolian children.

In addition, our work may be extended to model the adaptations to vitamin D scarcity: for these races, the uptake of calcium from food passing through the intestines might be higher [32,33,34,35]; the rate of conversion of vitamin D to its active form (i.e., from 25(OH)D to 1,25(OH)2D) might be higher [33]; the binding of vitamin D to carrier proteins in the bloodstream might be stronger [36,37].

To evaluate specific scenarios, we may also need to customise our model to consider cultural adaptations. For instance, meat may be consumed only in a raw or boiled state, thus preserving a co-factor that reduces the risk of rickets independently of the meat’s vitamin D content [38,39]. In some culture, breastfeeding of children continues for at least two years after birth. Mother’s milk is rich in beta-casein and other co-factors that increase the bioavailability of calcium [40,41,42]. Hence, our model needs to be extended to answer these specific questions.

## 5. Conclusions

Here, we developed a population PBPK model to successfully fit the sparsely sampled 3-year time course of serum 25(OH)D concentrations in healthy Cape Town children started between 6 and 11 years. This model accurately predicted serum 25(OH)D for up to 3 years in 463 children from the same study who were not used to fit the model. Model simulation results agreed with Cashman’s finding that 24.4 µg daily vitamin D is needed to achieve the 50 nmol/L sufficiency threshold in over 97.5% European children. The inferred volume of distribution was in agreement with another study on children with chronic kidney disease [12]. This model may be used to generate long-term forecasts in Cape Town children. However, Ulaanbaatar children exhibited significant differences in 25(OH)D dose–PK relationship which warrants further investigation.

## Figures and Tables

**Figure 1 nutrients-17-03028-f001:**
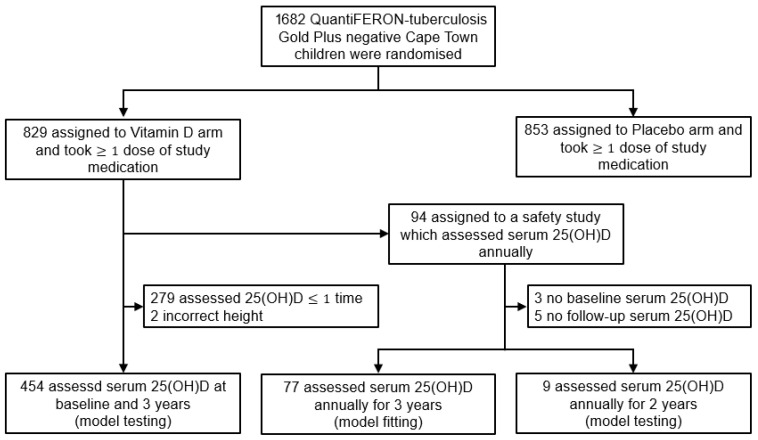
Flowchart of subject selection for model fitting and testing.

**Figure 2 nutrients-17-03028-f002:**
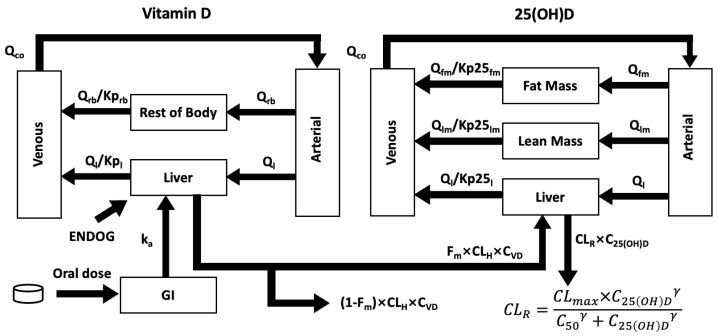
Schematic diagram of the final model. Following the gastrointestinal (GI) tract, vitamin D is metabolised in the liver and circulates to the rest of the body through venous blood and arterial blood. Vitamin D from endogenous synthesis and food intake (ENDOG) was incorporated into the model to account for a different baseline for each child. For 25(OH)D, non-eliminating organs were divided into lean mass and fat mass.

**Figure 3 nutrients-17-03028-f003:**
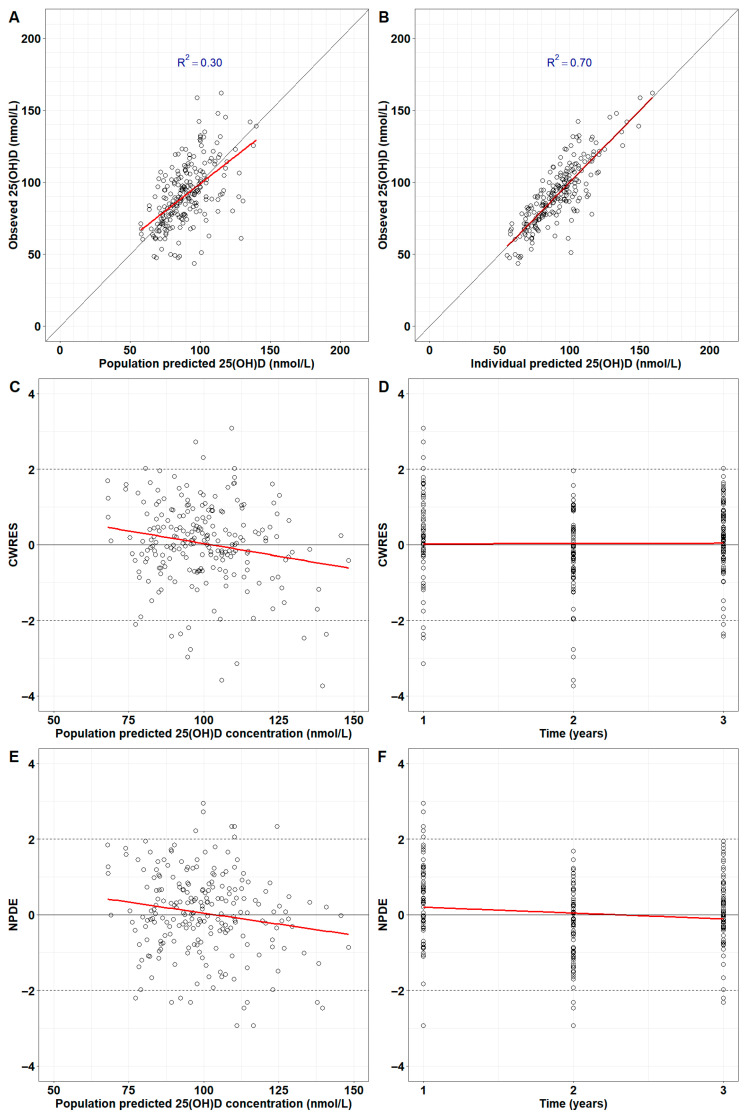
Diagnostic plots for the final model. Goodness-of-fit for population (**A**) and individual predictions (**B**) against observed serum 25(OH)D concentrations. Conditional weighted residuals (CWRES) were plotted against population predictions (**C**) and time (**D**). Normalised prediction distribution errors (NPDE) were plotted against population predictions (**E**) and time (**F**). Red lines: first-order linear regression.

**Figure 4 nutrients-17-03028-f004:**
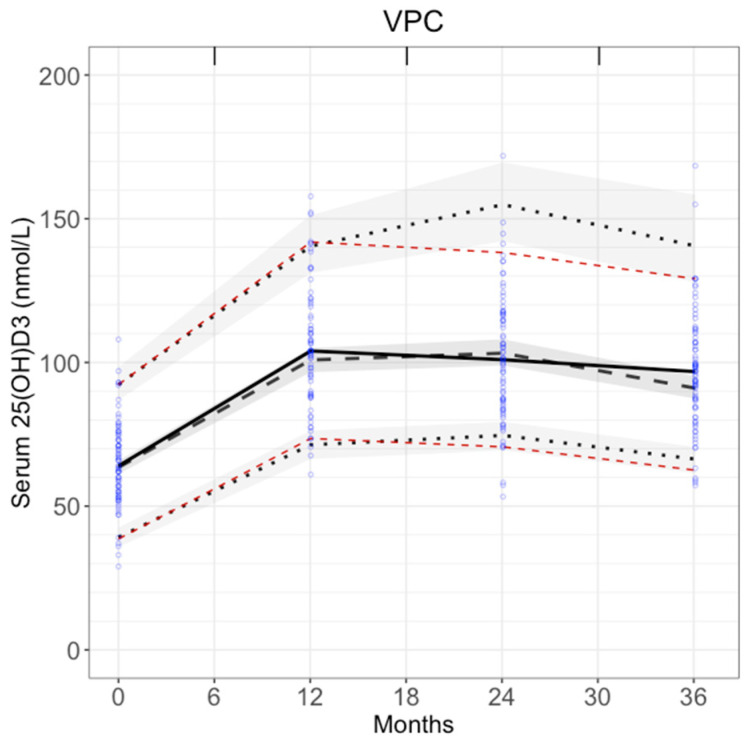
VPC of the final model. Model is simulated 1000 times. Blue dots: observations. Red dashed lines: observed 5th and 95th percentiles. Black dashed lines: simulated 5th and 95th percentiles. Grey shades around black dashed lines: 90% prediction intervals for the 5th and 95th simulated percentiles. Black solid line: median observations. Black dashed lines in bold: median predictions. Grey shades around median predictions: 90% prediction intervals for the median. Results are binned by each time point.

**Figure 5 nutrients-17-03028-f005:**
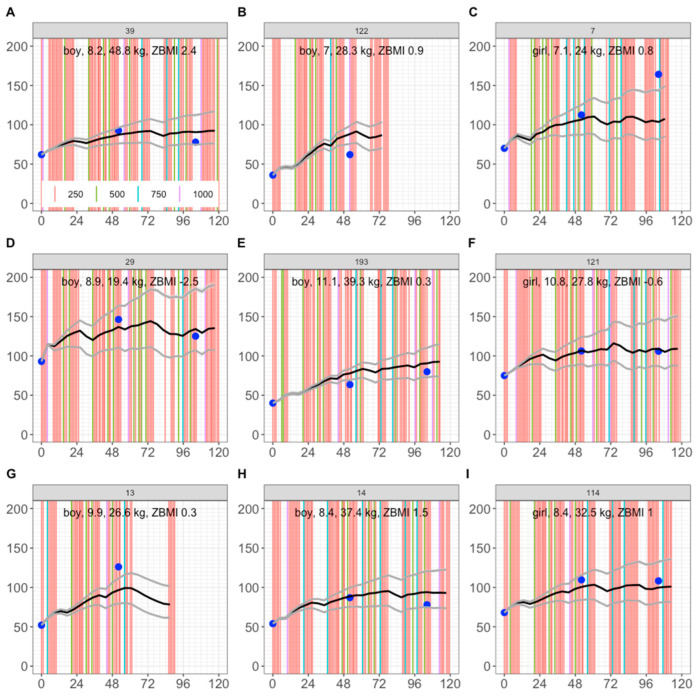
Simulation of serum 25(OH)D concentrations in 9 Cape Town children not used for model fitting. Horizonal axis: time in weeks. Vertical axis: Serum 25(OH)D in nmol/L. Blue dots: observations. Solid lines: 10th, 50th, and 90th percentiles. The 95% prediction intervals are too narrow to be seen. Vertical lines: dose in µg (salmon = 250; green = 500; cyan = 750; purple = 1000). In terms of subject ID: (**A**) 39; (**B**) 122; (**C**) 7; (**D**) 29; (**E**) 193; (**F**) 121; (**G**) 13; (**H**) 14; (**I**) 114.

**Figure 6 nutrients-17-03028-f006:**
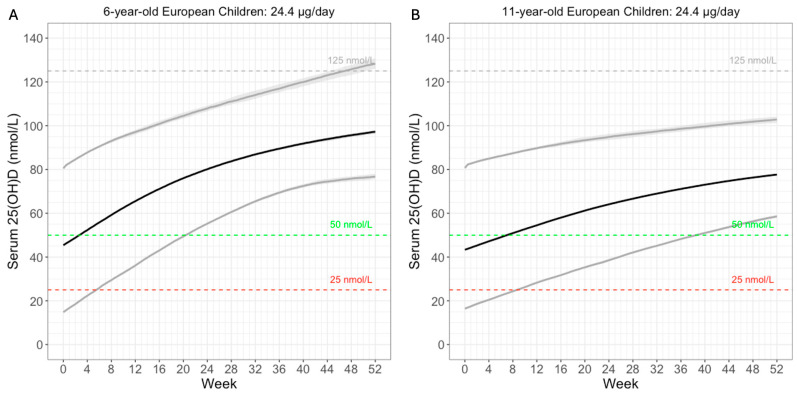
Simulated serum 25(OH)D PK at 24.4 µg vitamin D QD for one year. (**A**) 6 years. (**B**) 11 years. Lines: 2.5th, 50th, and 97.5th percentiles. Shaded area: 95% prediction intervals. Severe deficiency: 25 nmol/L. Deficiency: 50 nmol/L. Potential toxicity: 125 nmol/L.

**Table 1 nutrients-17-03028-t001:** Fixed and random effects in all models.

Model	Fixed Effects	Random Effects
1–8	${\mathrm{CL}}_{\max}$ $,\;K_{p25\mathrm{rb}}$	$\eta_{{\mathrm{CL}}_{\max}}$, $\eta_{K_{p25\mathrm{rb}}}$
9	${\mathrm{CL}}_{\max}$ $,\;K_{p25\mathrm{fm}}$	$\eta_{{\mathrm{CL}}_{\max}}$
10	${\mathrm{CL}}_{\max}$ $,\;K_{p25\mathrm{fm}}$ $,\;K_{p25\mathrm{lm}}$	$\eta_{{\mathrm{CL}}_{\max}}$
11	${\mathrm{CL}}_{\max}$ $,\;K_{p25\mathrm{rb}}$ $,\;{\mathrm{CL}}_{C50}$	$\eta_{{\mathrm{CL}}_{\max}}$, $\eta_{K_{p25\mathrm{rb}}}$

**Table 2 nutrients-17-03028-t002:** Parameter estimates for the final model.

	Est. in Natural Log	SE in Natural Log	% RSE	Linear Scale (95% CI)	IIV % CV *	Shrink %
CL_max_ (h^−1^)	−4.43	0.0787	1.78	0.0119 (0.0102, 0.0139)	62.8	13.1%
Kp_25fm_	1.54	0.136	8.8	4.66 (3.6, 6.11)		
Additive error (nmol/L)	0.00249			0.00249		
Proportional error	0.109			0.109		

* IIV CV% is $\sqrt{e^{\omega }-1}\times 100\%$, where ω is the variance of random effects. Hence, $\omega_{{CL}_{max}}=0.332$.

## Data Availability

Anonymized data from the ViDiKids Cape Town Trial may be requested from Prof Adrian Martineau to be shared subject to terms of research ethics committee approval. Data requests for the ViDiKids Mongolian Trial should be sent to Dr Davaasambuu Ganmaa (gdavaasa@hsph.harvard.edu) and/or Prof Adrian Martineau (a.martineau@qmul.ac.uk) subject to terms of research ethics committee approval.

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
