# Peer review of "Physiologically Based Pharmacokinetic Modelling of Serum 25-Hydroxyvitamin D Concentrations in Schoolchildren Receiving Weekly Oral Vitamin D3 Supplementation"

_nutrients, 2025, doi:10.3390/nu17193028_

Round 1
Reviewer 1 Report
Comments and Suggestions for Authors
The paper “Physiologically-based Pharmacokinetic Modelling of Serum 25-hydroxyvitamin D Concentrations in Schoolchildren Receiving Weekly Oral Vitamin D3 Supplementation” contributes to the growth of literature for research on the area of vitamin D supplementation, especially among children.
However, before the manuscript is accepted for publication in “Nutrients” the following items should be revised:
I suggest clearly presenting the purpose of the work. Developing hypotheses would enhance the scientific value of the work.
Methodology
What was the method of recruiting the study group?
This concerns the comparison of results from studies conducted among Mongolian children. Why is the research group and recruitment method not described in detail?
Were there any similar breaks in the research?
Reviewer 2 Report
Comments and Suggestions for Authors
The authors should include within their very short discussion further comparison of their study model and results with the results and methods of other studies on the same subject. The authors should include within their discussion the limitations of their study. The authors should include within their study further details on the scientific basis of applying simulation data in the scientific comparisons of their study. The authors should expand on the necessity and utility of their piece of research. The authors should expand on the scientific basis of their model of assessment of vitamin D levels after vitamin D supplementation. The authors should expand in the discussion in the comparison of their findings with findings of other authors.
Comments on the Quality of English LanguageThe use of the English language needs some improvement.
Reviewer 3 Report
Comments and Suggestions for Authors I compliment the design and conduct of the study.I have only two comments: the first concerns the possible role of other vitamin D
metabolites in assessing biologically active levels.
The second concerns the influence of renal function,
if and how it was included in the model.
Reviewer 4 Report
Comments and Suggestions for Authors
The most intriguing finding is the apparent difference in vitamin D metabolism between the Cape Town and Mongolian participants. Despite lower compliance and a lower weekly dose, serum 25(OH)D showed a greater increase in Cape Town children on average than in Mongolian children. In fact, many of the Mongolian children reported a 3-year serum 25(OH) that was lower than the baseline.
There is a homeostatic mechanism that resists increases in serum 25(OH)D above a certain set-point. This set-point seems to vary among human populations, being lower in those that live at high latitudes or have very dark skin. In both cases, vitamin D is produced less easily in the skin, and humans have adapted by using it more sparingly.
The existence of a homeostatic mechanism is shown by a study of African Americans with varying degrees of African ancestry. Both sunlight and diet were 46% less effective in raising the 25(OH)D levels of participants with high African ancestry, as compared to those with low African ancestry (Signorello et al., 2010).
Adaptation to low serum 25(OH)D has been shown by several studies of high-latitude populations, mainly the Inuit of Canada and Greenland but also indigenous populations of northern Asia (Frost, 2012; Frost, 2018). Like the Inuit, Mongolians have adapted to an environment where the skin cannot produce sufficient vitamin D for most of the year.
Adaptations to vitamin D scarcity include certain physiological changes:
- Higher uptake of calcium from food passing through the intestines (Sellers et al., 2003; also see Rejnmark et al., 2004; Skjøth et al., 2025; Waiters et al., 1999).
- Higher rate of conversion of vitamin D to its most active form (i.e., from 25(OH)D to 1,25(OH)2D) (Rejnmark et al., 2004).
- Stronger binding of vitamin D to carrier proteins in the bloodstream (Larcombe et al., 2012; Malyarchuk, 2020).
There are also cultural adaptations:
- Consumption of meat only in a raw or boiled state, thus preserving a co-factor that reduces the risk of rickets independently of the meat’s vitamin D content (Dunnigan et al., 2005; Mellanby, 1918).
- Breastfeeding of children for at least two years after birth. Mother’s milk is rich in beta-casein and other co-factors that increase the bioavailability of calcium (Frost, 2022; Kent et al., 2009; Lönnerdal, 2003).
In my opinion, the authors should mention the literature on population differences in vitamin D metabolism, particularly the Signorello et al. (2010) study.
References
Dunnigan, M.G., Henderson, J.B., Hole, D.J., Mawer, E.B., & Berry, J.L. (2005). Meat Consumption Reduces the Risk of Nutritional Rickets and Osteomalacia. Brit. J. Nutr. 94, 983-991. https://doi.org/10.1079/BJN20051558
Frost, P. (2012). Vitamin D deficiency among northern Native Peoples: a real or apparent problem? International Journal of Circumpolar Health, 71(S2), 18001. https://doi.org/10.3402/IJCH.v71i0.18001
Frost, P. (2018). To supplement or not to supplement: are Inuit getting enough vitamin D? Études Inuit Studies, 40(2), 271-291. https://doi.org/10.7202/1055442ar
Frost P. (2022) The Problem of Vitamin D Scarcity: Cultural and Genetic Solutions by Indigenous Arctic and Tropical Peoples. Nutrients, 14(19), 4071. https://doi.org/10.3390/nu14194071
Kent, J.C., Arthur, P.G., Mitoulas, L.R., & Hartmann, P.E. (2009). Why calcium in breastmilk is independent of maternal dietary calcium and vitamin D. Breastfeeding Review, 17, 5-11.
Larcombe, L., Mookherjee, N., Slater, J., Slivinski, C., Singer, M., Whaley, C., Denechezhe, L., Matyas, S., Turner-Brannen, E., Nickerson, P., & Orr, P. (2012). Vitamin D in a Northern Canadian First Nation Population: Dietary Intake, Serum Concentrations and Functional Gene Polymorphisms. PLoS ONE, 7(11): e49872. https://doi.org/10.1371/journal.pone.0049872
Lönnerdal, B. (2003). Nutritional and Physiologic Significance of Human Milk Proteins. American Journal of Clinical Nutrition, 77, 1537S-1543S. https://doi.org/10.1093/ajcn/77.6.1537S
Malyarchuk, B.A. (2020). Polymorphism of GC gene, encoding vitamin D binding protein, in aboriginal populations of Siberia. Ecological Genetics, 18, 243-250. https://doi.org/10.17816/ecogen18634
Mellanby, E. (1918). The part played by an ‘accessory factor’ in the production of experimental rickets. Proceedings of the Physiological Society. xi–xii.
Rejnmark, L., Jørgensen, M.E., Pedersen, M.B., Hansen, J.C., Heickendorff, L., Lauridsen, A.L., Mulvad, G., Siggaard, C., Skjoldborg, H., Sørensen, T.B., et al. (2004). Vitamin D insufficiency in Greenlanders on a Westernized fare: ethnic differences in calcitropic hormones between Greenlanders and Danes. Calcified Tissue International, 74, 255–263. https://doi.org/10.1007/s00223-003-0110-9
Sellers, E.A.C., Sharma, A. & Rodd, C. (2003). Adaptation of Inuit children to a low-calcium diet. Canadian Medical Association Journal, 168(9), 1141–1143.
Signorello, L.B., Williams, S.M., Zheng, W., Smith, J.R., Long, J., Cai, Q., Hargreaves, M.K., Hollis, B.W., & Blot, W.J. (2010). Blood vitamin D levels in relation to genetic estimation of African ancestry. Cancer Epidemiology, Biomarkers & Prevention, 19, 2325–2331. https://doi.org/10.1158/1055-9965.EPI-10-0482
Skjøth, J. B., Hagens, T. M., Fleischer, I., Laursen, M., & Andersen, S. (2025). Bone mineral content among Inuit – a systematic review of data. International Journal of Circumpolar Health, 84(1). https://doi.org/10.1080/22423982.2025.2502249
Waiters, B., Godel, J.C., & Basu, T.K. (1999). Perinatal Vitamin D and Calcium Status of Northern Canadian Mothers and their Newborn Infants. Journal of the American College of Nutrition, 18, 122-126. https://doi.org/10.1080/07315724.1999.10718839
